

# A robust numerical method for the generation and simulation of periodic finite-amplitude internal waves in natural waters

Pierre Lloret[1], Peter J. Diamessis[1], Marek Stastna[2], and Greg N. Thomsen[3]

[1]School of Civil and Environmental Engineering, Cornell University, Ithaca, NY, USA
[2]Department of Applied Mathematics, University of Waterloo, Waterloo, ON, Canada
[3]Wandering Wakhs Research, Austin, TX, USA

**Correspondence:** Pierre Lloret (pel62@cornell.edu)

**Abstract.** The design and implementation of boundary conditions for the robust generation and simulation of periodic finite-amplitude internal waves is examined in a quasi two-layer continuous stratification using a spectral-element-method-based incompressible flow solver. The commonly-used Eulerian approach develops spurious, and potentially catastrophic, small-scale numerical features near the wave-generating boundary in a nonlinear stratification when the parameter $A/(\delta\ c)$ is sufficiently

larger than unity ; $A$, $\delta$ are measures of the maximum wave-induced vertical velocity and pycnocline thickness, respectively, and $c$ is the linear wave propagation speed. To this end, an Euler-Lagrange approach is developed and implemented to generate robust high-amplitude periodic deep-water internal waves. Central to this approach is to take into account the wave-induced (isopycnal) displacement of the pycnocline in both the vertical and (effectively) upstream directions. With amplitudes not restricted by the limits of linear theory, the Euler-Lagrange-generated waves maintain their structural integrity as they propagate

away from the source. The advantages of the high-accuracy numerical method, whose minimal numerical dissipation cannot damp the above near-source spurious numerical features of the purely Eulerian case, can still be preserved and leveraged further along the wave propagation path through the robust reproduction of the nonlinear adjustments of the waveform. The near-and-far-source robustness of the optimized Euler-Lagrange approach is demonstrated for finite-amplitude waves in a sharp quasi two-layer continuous stratification representative of seasonally stratified lakes. The findings of this study provide an enabling

framework for two-dimensional simulations of internal swash zones driven by well-developed nonlinear internal waves and, ultimately, the accompanying turbulence-resolving three-dimensional simulations.

## 1 Introduction

Internal swash zones (ISZs) (Emery and Gunnerson, 1973; Woodson, 2018) are regions which develop along sloping oceanic boundaries through the action of periodically incident internal waves (IWs) in a manner analogous to a surface swash zone

on the beach, albeit at slower timescales (O(10 minutes) or longer) and over longer wavelengths (O(1 km) or longer) (Cowen et al., 2003; Elfrink and Baldock, 2002). In ISZs, energy can flux downscale to turbulence effectively either through shear or convective instabilities in the IW interior, similarly to spilling or plunging breaker waves on the ocean surface (Cowen et al., 2003; Ting and Kirby, 1996), or through the turbulent boundary layer established through the interaction of the IW-induced current with the seafloor (Zulberti et al., 2022). In the latter context, particularly strong turbulence can be generated in the form





of a near-bottom turbulent wake due to boundary layer separation associated with the along-bed wave-induced adverse pressure gradient induced by either internal bores or internal solitary waves (Hosegood et al., 2004; Boegman and Stastna, 2019). The above turbulence-generation mechanisms presumably conspire to drive significant boundary-interior exchange (McPhee-Shaw and Kunze, 2002; McPhee-Shaw, 2006), i.e., the exchange of water between the boundary layer and the stratified interior, which effectively drives mixing in the relatively less active stratified water-body interior (McPhee-Shaw and Kunze, 2002;

Boegman and Ivey, 2009). The periodic shoaling and breaking of the above IWs in shallow environments, like continental shelves or slopes, has a direct impact on the internal thermal equilibrium and biogeochemistry of the water column (Woodson, 2018). The periodically on-slope incident IWs are important in the transfer of mass, whether transporting nutrients and plankton toward the surface in the inner shelf (Omand et al., 2015) or whether ejecting bottom boundary layer sediments as high as 40 meters into the water column during a strong vertical updraft event associated with the passage of a nonlinear internal wave of

depression over the slope (Cheriton et al., 2016). A similar class of long IW-driven phenomena, of comparable biogeochemical importance, also occur on the slopes of lakes (Thorpe, 1998; Wuest and Lorke, 2003) and have served as the primary motivator of the research presented here.

The leading-order component of the periodic wave field forcing of an ISZ consists of lower vertical mode IWs whose wavelength is $O(50 - 100)$ longer than the water column depth, namely in the form of oceanic internal tidal waves or the

basin-scale internal seiche of a lake (Emery and Gunnerson, 1973; Nash et al., 2004; Stevens et al., 2005; Martini et al., 2013; Lemckert and Imberger, 1998). In the latter case, the internal seiche is further associated with a lower horizontal mode associated with the longer dimension of the lake. Such long waves are commonly expected to be represented, with sufficient fidelity, through the use of linear IW theory (Stastna, 2022) at, nonetheless, values of *finite wave amplitude*. Frequently, higher-frequency/shorter-wavelength highly nonlinear features, such as turbulent/undular bores or internal solitary waves (Stastna,

2022) may be embedded within the longer incident IW (Hosegood et al., 2004; Lucas and Pinkel, 2022; Thorpe et al., 1996).

The primary objective of this paper is the development of a robust numerical method for the generation and subsequent development of the longer component of the deep-water wave-forcing at *finite amplitude*. The generated wave should have an amplitude that is not constrained by the limits of linear theory. Practically, this corresponds to wave-induced maximum isopycnal displacements that are at least $5\%$ of the total water column depth. The wave should also remain sufficiently *robust*

*near the source* and, with an equal degree of robustness, *nonlinearly adjustment of its waveform* as it propagates along the waveguide. In this regard, central to this paper's scope is that the background stratification extend beyond an uniform density gradient (Taylor, 1993) and is actually subject to variation in the vertical, as characterized by the presence of a distinct pycno-cline which is commonly a close approximation of the in-situ background profiles in the stratified ocean or lakes. Finally, an additional essential ingredient of this study, is that a high-accuracy discretization is used, specifically a nodal spectral element

method (Diamantopoulos et al., 2022). The particular discretization technique enables the optimal resolution of the generated waves, their nonlinear adjustments away from the source and ultimately (though not explicitly considered here) the associated instabilities/turbulence upon encounter of the waves with the slope.

In the laboratory, one approach to generate periodic long internal IWs is by tilting and releasing the actual laboratory tank (Boegman et al., 2005): the resulting horizontal standing wave is a lab-scale surrogate of the basin-scale internal seiche



generated in a long stratified lake in response to a strong wind event (Boegman, 2009). An equivalent type of horizontal standing wave may be generated in a numerical simulation within a long rectangular computational domain by using an initial condition consisting of a tilted pycnocline (Grace et al., 2019). One issue with the tilting-based wave-generation approach may be that it immediately produces finite velocities across the whole domain/tank, when one would prefer waves propagating into an initially quiescent slope region. In many cases, the standing wave will breakdown into a propagating wave train (Grace

et al., 2019).

An alternative, more flexible and effectively more controllable, wave generation approach involves introducing a form deep-water (far from the slope) oscillatory wave-excitation. Such an approach would ideally allow a *sufficiently long propagation distance in uniform depth waters*, prior to the wave encountering the slope, which permits the generated IW to undergo any required nonlinear adjustments. To this end, in the deep-water section of a laboratory tank, a horizontally oscillating paddle

(Wallace and Wilkinson, 1988; Nakayama and Imberger, 2010; Ghassemi et al., 2022), a vertically oscillating semi-cylinder (Moore et al., 2016) or an array of plates vertically stacked on an eccentric camshaft (Mercier et al., 2010, 2013) have been used. It is worth noting that all the above experimental studies generated relatively short waves, as represented by values of aspect ratio $\lambda/H$ and non-dimensional amplitude $\eta_{max}/H$ ; $\lambda$, $\eta_{max}$ and $H$ are the IW horizontal wavelength, IW-induced maximum isopycnal displacement and water depth, respectively. Reported directly, or inferred, values of $\lambda/H$ and $\eta_{max}/H$ lie

in the range [2, 12.5] and [0.0075, 0.2] respectively, in the above laboratory studies wherever directly identifiable or inferable. Such a maximum pycnocline displacement range corresponds to a wave Froude number, $Fr = U_{max}/c$, of [0.02, 0.35]; $U_{max}$ and $c$ are the maximum wave-induced horizontal current and wave propagation speed.

The high-order-accuracy turbulence-resolving fully nonlinear and non-hydrostatic three-dimensional simulations of Winters (2015) are one of the few computational studies so far which has considered the generation and incidence of a periodic long

wave and on a, relatively steep, slope. The wave aspect ratio and wave Froude number can be inferred as $\lambda/H = 48$ and $Fr = 0.1385$. Note that the work of Winters considered only a uniform background stratification. Moreover, his generated waves were allowed a distance less than one prescribed wavelength from the source to propagate until the slope most likely precluding any deep-water nonlinear adjustments of the waveform.

To the authors best knowledge, the only other computational study which has examined the periodic generation, the prop-

agation away from the source over at least one wavelength and the incidence of long internal waves on a slope is the two-dimensional investigation by Dauhajre et al. (2021). The wave aspect ratios considered in this study are high and can be inferred as residing in the range [50, 400], while noting the very small ratio of computational domain depth to length. The wave based Froude number values considered are in the range [0.05, 0.4]. Furthermore, the subset of simulations that use a two-layer stratification (and not a linear one) have a thick pycnocline and focus on an aspect ratio of $\lambda/H = 200$ and a wave

based Froude number between 0.1 and 0.2. Note also the curvature of the density profile at the base of the pycnocline is reduced by introducing a weakly stratified layer below. The numerical dissipation inherently built into the parameterizations of the regional ocean modeling code (ROMS) used in this study could also effectively damp any spurious numerical features near the wave-generating deep-water boundary.



Note that the studies of Masunaga et al. (2015, 2016); Walter et al. (2012) also considered a non-uniform stratification but
positioned the wave-generating source only a fraction of the target wavelength from the slope. The generated waves, therefore,
were not afforded an adequate propagation distance to undergo any nonlinear adjustments before encountering the slope.
Additionally, per this paper's focus on periodic IW simulation with high-accuracy methods and high resolution, the nesting-
based robust mode-1 long internal tide generation within a regional-scale nonhydrostatic model (Rogers et al., 2019) is not
pertinent to the scope of this study as it relies on low-pass filtering and sponge layers.

The laboratory and computational studies discussed above consider generated waves that may be deemed as either short
or long. Even when high-accuracy/resolution numerical methods are derived and efficiently implemented on a state-of-the-art
high-performance computing platform, a computational study aiming to sufficiently resolve instability/turbulence formation
due to sufficiently high-amplitude waves over a limited number cycles of an ISZ is practically limited to a wave aspect ratio in
the range [40, 50]. This is the aspect ratio regime accessed by the work of Winters (Winters, 2015), which is however limited
to a linear stratification. The choice of a linear stratification effectively shielded this study from the challenges that emerge
when forcing internal waves in a pycnocline–dominated stratification profile. As will be demonstrated later in this paper, the
generation of high-amplitude periodic internal waves in more general, nonlinear, stratifications for waves operating in this
intermediate aspect ratio range is confronted with non-trivial error if commonly used deep-water forcing approaches, such as
those employed by Winters and Dauhajre et al., are actually employed. The minimal numerical dissipation of a high-order-
accuracy numerical method can allow this error to grow substantially. The stability of the simulation can thus be effectively
undermined, and one can no longer leverage the high-accuracy of the method for representing nonlinear wave adjustments in
deeper-water and the finer-scale features once the slope is reached.

Thus from a computational point of view a relatively simple technique for generating larger-amplitude IWs for general
stratifications in deep water is highly desirable. This is often achieved by choosing a form of boundary conditions at the
boundary away from the slope region. For most field-relevant stratifications, a pycnocline dominates the stratification, and
the vertical motion of then pycnocline is the clearest manifestation of internal waves. Historically, descriptions of internal
waves typically built on a linearized theory and the literature has examples of two different choices for the vertical coordinate:
one which uses the physical coordinate $z$, and one which uses the upstream height of each isopycnal (and more concretely,
the upstream height of the dominant pycnocline) (Gear and Grimshaw, 1983; Yih, 1977). Since the former uses the physical
coordinates, it is usually labelled as the *Eulerian* theory of linear internal waves. The latter, in contrast, is labelled the *Euler-
Lagrange* theory because the horizontal coordinate is the physical coordinate $x$, while the vertical coordinate is the upstream
coordinate, often written as $y = z - \eta(x, z, t)$ where $\eta$ is the the isopycnal displacement. Both the Eulerian (Lamb and Yan,
1996) and Euler-Lagrange theories (Gear and Grimshaw, 1983) have been used as a basis for multi-scale asymptotic expansions
that extend the wave description to small, but finite amplitude waves (i.e. weak nonlinearity), and waves of finite wavelength
(i.e. weak dispersion). These lead to model equations in the Korteweg-de-Vries family. The use of the upstream isopycnal
height has found general use in the description of stratified flow, in both the classical (Yih, 1977) and modern (Stastna, 2022)
contexts. In the simulation context, the desire to generate finite amplitude waves in a situation with a strong pycnocline implies





that forcing methodologies based on Eulerian, linear wave theory may not yield robust results. The Euler-Lagrange theory offers an alternative, if algebraically more complex, development pathway.

In this paper, by following an Eulerian and an Euler-Lagrange approach (Turkington et al., 1991; Gear and Grimshaw, 1983), different types of time-dependent periodic wave-generating boundary conditions are derived with a particular emphasis on the subtleties associated with a continuous two-layer background stratification. The efficacy of each approach in generating a robust deep-water periodic finite-amplitude IW train and enabling any nonlinear adjustments of the wave-train is thereafter assessed.

## 2 Problem set-up and model formulation

### 2.1 Problem geometry

The canonical flow examined in this paper is the propagation of a two-dimensional finite amplitude periodically forced internal wave in a quasi two-layer continuous stratification. The computational domain is a two-dimensional rectangle of dimensions $L \times H$ and is stratified in the vertical direction $z$ with a vertically varying buoyancy frequency $N(z)$ where

$$N^2(z) \equiv -\frac{g}{\rho_0}\frac{d\overline{\rho}}{dz}. \tag{1}$$

Restricting one's focus to the Boussinesq approximation, the total density is decomposed as the addition of a reference density $\rho_0$, a stratification $\overline{\rho}$ and a perturbation $\rho'$ (Kundu et al., 2008):

$$\rho(\boldsymbol{x},t) = \rho_0 + \bar{\rho}(z) + \rho'(\boldsymbol{x},t) \text{ with } \rho' \ll \bar{\rho} \ll \rho_0. \tag{2}$$

The quasi two-layer continuous stratification $\overline{\rho}$ (Fig. 1, left panel) is defined by:

$$\overline{\rho}(z) = -\frac{\rho_0 N_0^2 \delta}{g} tanh\left(\frac{z - z_p}{\delta}\right), \tag{3}$$

where $\rho_0 N_0^2 \delta/g$ is a measure of the density difference across the pycnocline, $N_0$ is a reference buoyancy frequency equal to the peak value of $N(z)$ in the water column, $\delta$ is a measure of the pycnocline thickness and $z_p$ is the position of the pycnocline's center.

The finite amplitude internal wave, of wavenumber $k$ and angular frequency $\omega$, is generated through a forcing implemented

within the left boundary conditions. Details on the exact derivation of the deep water boundary conditions will be covered in Section 3.

### 2.2 Governing equations

The governing equations for the problem are the incompressible Navier-Stokes equations (INSE) under the Boussinesq approximation, written as:





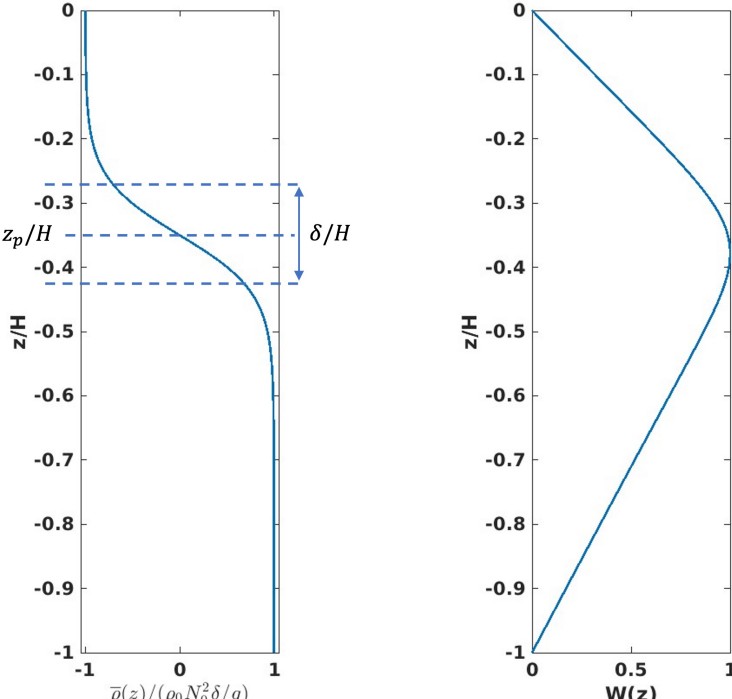

**Figure 1. Left**: A two layer continuous stratification $\overline{\rho}$, defined by its density jump $2\rho_0 \frac{N_0^2 \delta}{g}$, its position $z_p$ and its thickness $\delta$. **Right**: Corresponding vertical structure eigenfunction $W(z)$ which is the solution to the eigenvalue problem Eq. (11).

.

$$\frac{\partial \mathbf{u}}{\partial t} = -\mathbf{u} \cdot \nabla \mathbf{u} - \frac{g}{\rho_0} \rho' \mathbf{k} - \frac{1}{\rho_0} \nabla p' + \nu \nabla^2 \mathbf{u} \,, \tag{4}$$

$$\frac{\partial \rho'}{\partial t} = -\mathbf{u} \cdot \nabla \rho + \kappa \nabla^2 \rho' \,, \tag{5}$$

$$\nabla \cdot \mathbf{u} = 0 \,. \tag{6}$$

The simulations reported here will be limited to two dimensions on the $(x, z)$ plane. Therefore $\mathbf{u}$ will be limited to its two components $(u, w)$. Furthermore $\rho'$ is the density perturbation as defined in Eq. (2) and $p'$ is the pressure perturbation, defined as the deviation from the hydrostatic pressure. It is important to note that the hydrostatic balance between $\overline{\rho}(z)$ and the corresponding background pressure field has been subtracted from Eq. (5). Here $\mathbf{k}$ is the unit vector in the positive direction, $\nu$ and $\kappa$ are the constant kinematic viscosity and mass diffusivity and $g$ is the gravitational constant.





### 2.3 Wave-based dimensionless parameters

The definition of the wave-based Reynolds number $Re_w$ is given by

$$Re_w \equiv \frac{\lambda c}{\nu}, \tag{7}$$

where $\lambda$ is the associated wavelength and $c = \omega/k$ the wave-speed of the prescribed wave. The particular wave-based Reynolds number is independent of the wave amplitude and quantifies the strength of viscous effects during the time required for the wave to propagate a distance of one wavelength $\lambda$.

The wave aspect ratio $\lambda/H$ is a metric used to describe how long the wave is relatively to the water depth. Additionally, in the context of this study, it effectively represents the upstream variation of wave-induced flow fields at the wave-generating deep-water boundary which then determines which variant of an Euler-Lagrange-approach must be used.

The Froude number quantifies the strength of nonlinear effects within the wave against the restoring effect of buoyancy and is defined as:

$$Fr \equiv \frac{U_0}{c}, \tag{8}$$

where $U_0$ is the maximum-wave induced horizontal fluid velocity.

### 2.4 Numerical method

The numerical method used to generate the simulation data sets examined in this paper is a high-order continuous-Galerkin numerical method, originally developed for the simulation of non-linear, non-hydrostatic internal waves and turbulence in long computational domains with complex bathymetry. Details on the numerical model may be found elsewhere (Diamantopoulos et al., 2022). The time discretization is semi-implicit and relies on a third order stiffly stable scheme (Karniadakis et al., 1991). The spatial discretization is based on the nodal spectral element method. Such a discretization enables robust wave propagation against numerical dispersion and diffusion effects, a highly accurate representation of complex geometries and a flexibility in localized resolution, namely the across pycnocline.

Except for the deep water boundary conditions, a free-slip boundary condition is prescribed for the velocity field along all other boundaries. The density is subject to a boundary condition of zero diffusive mass flux along all domain boundaries:

$$\frac{\partial \rho}{\partial n} = \nabla \rho \cdot \mathbf{n} = 0. \tag{9}$$

## 3   Deep water wave-generating boundary conditions

The generation of finite-amplitude periodic internal waves is a key component of this study. To this end, we examine the spatio-temporal structure of the generated wave, as prescribed by linear theory, which is introduced into the computational domain in the form of time-dependent, vertically-variable Dirichlet conditions at the deep-water boundary.



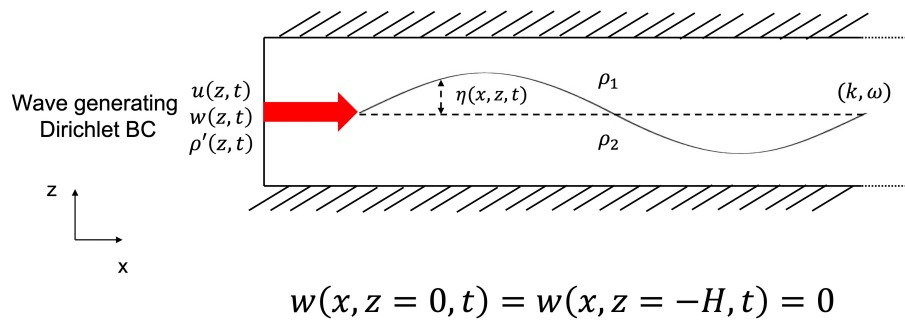

$$w(x, z = 0, t) = w(x, z = -H, t) = 0$$

**Figure 2.** Schematic of the generation of a mode-1 wave in the wave guide analogy using time-dependent boundary conditions for a two-layer stratification, highlighting the pycnocline displacement $\eta$.

## 3.1 Internal wave vertical structure: mathematical descriptions

The fluid's top and bottom boundaries naturally confine the propagation of internal waves so that it occurs in the horizontal direction, along a waveguide formed by the naturally occurring density stratification as shown in Fig. 2. This density stratification, which only varies in the $z$ direction, is often dominated by a region of rapid change (the so–called pycnocline) and it is the up and down motion of this pycnocline that is essential for an accurate description of wave motion.

Mathematically, internal waves can be represented as a separation of variables solution with a fixed, or standing wave, structure in the vertical and a propagating waveform (a plane wave in the linear theory) in the horizontal. If we choose the propagation direction to be from left to right, and assume the waves to be periodic in $x$ and $t$, the vertical component of velocity will have the form:

$$w(\boldsymbol{x}, t) = W(z) exp\left[i(kx - \omega t)\right]. \tag{10}$$

The equation governing the vertical structure may be derived by linearizing the stratified Euler equations under the Boussinesq approximation (which in turn result from dropping the viscous/diffusive terms in Eq. (4)), performing a series of algebraic manipulations to leave an equation for $w$ only, and introducing the wave ansatz above. The buoyancy frequency profile $N(z)$ is assumed given, and for the scope of this paper we are neglecting any form of background shear current. $W(z)$ then becomes the solution of the following linear eigenvalue problem (Gerkema and Zimmerman, 2008), for either $\omega$ or $k$ with the other parameter assumed specified:

$$\frac{d^2 W}{dz^2} + k^2 \frac{N^2(z) - \omega^2}{\omega^2} W = 0. \tag{11}$$

Both top and bottom boundaries are assumed impermeable such that:

$$W(0) = W(-H) = 0. \tag{12}$$





For a given wave number $k$, an infinite number of eigenfunctions $W_n(z)$ with their corresponding eigenvalues $\omega_n$ exist, each one representing a different vertical mode (i.e. mode–1 does not cross zero in the interior of the fluid, mode–2 crosses zero once in the interior of the fluid, etc). Therefore the general solution $w(\boldsymbol{x},t)$ can be represented by a superposition of such modes, using arbitrary constants $a_n \in \mathbb{C}$:

$$w(\boldsymbol{x},t) = \sum_n W_n(z)\left[a_n \exp(i(kx - \omega_n t))\right]. \tag{13}$$

Details regarding the derivation of the solution in the linear stratification case, which are pertinent to the discussion in Section 4, are provided in Appendix A.

### 3.2 Eulerian approach

The above description computes $w$ as a function of a *fixed coordinate system*. This is often called the "lab frame" and the theory is labelled as *Eulerian* (Kundu et al., 2008). In this first approach to generate a finite amplitude periodic IW, a two-dimensional perturbation field $(u_E, w_E, \rho'_E)$ is constructed from the solution of the eigenvalue problem for $W(z)$ via the following set of manipulations of the linearized, stratified Euler equations under the Boussinesq approximation:

$$\frac{\partial \mathbf{u_E}}{\partial t} = -\frac{g}{\rho_0}\rho'_E \mathbf{k} - \frac{1}{\rho_0}\nabla p', \tag{14}$$

$$\frac{\partial \rho'_E}{\partial t} = -w_E \frac{\partial \bar{\rho}}{\partial z}, \tag{15}$$

$$\nabla \cdot \mathbf{u_E} = 0 \,. \tag{16}$$

As mentioned above, such an approach can be considered as Eulerian since we are looking at the evolution in time of the wave induced velocity and density fields from a a fixed frame of reference. Without a loss of generality, since the chosen equations are linear, only a mode–1 wave will be considered, corresponding to the smallest wave number possible. Following the stratified waveguide analogy (see Section 3.1), and multiplying the result by an arbitrary scaling factor $A$, effectively a measure of wave amplitude, the resulting $w_E$ perturbation is:

$$w_E(x,z,t) = -Ak\cos(kx - \omega t)W(z). \tag{17}$$

Using continuity, Eq. (16), an expression for $u_E$ is derived accordingly:

$$u_E(x,z,t) = A\sin(kx - \omega t)\frac{dW}{dz}. \tag{18}$$

 

Further appealing to the linearized form of the advection-diffusion equation, Eq. (15), the density perturbation $\rho'_E$ is then:

$$\rho'_E(x,z,t) = -\frac{d\bar{\rho}}{dz}\frac{Ak}{\omega}\sin(kx - \omega t)W. \tag{19}$$

The result of the above derivation is a field $(u_E, w_E, \rho'_E)$, shown in Figs. 3 (a) and 3 (b) at an arbitrary time, of a propagating internal wave solution of the linear Euler equations under the Boussinesq approximation. The approximate fields are two-dimensional in space, $x, z$, and also depend on time $t$. They exhibit a separable structure in $x - t$ and the vertical direction $z$. In practice, the approximations are implemented through a Dirichlet boundary condition along a vertical boundary, which we assume to occur at $x = 0$, without any loss of generality.

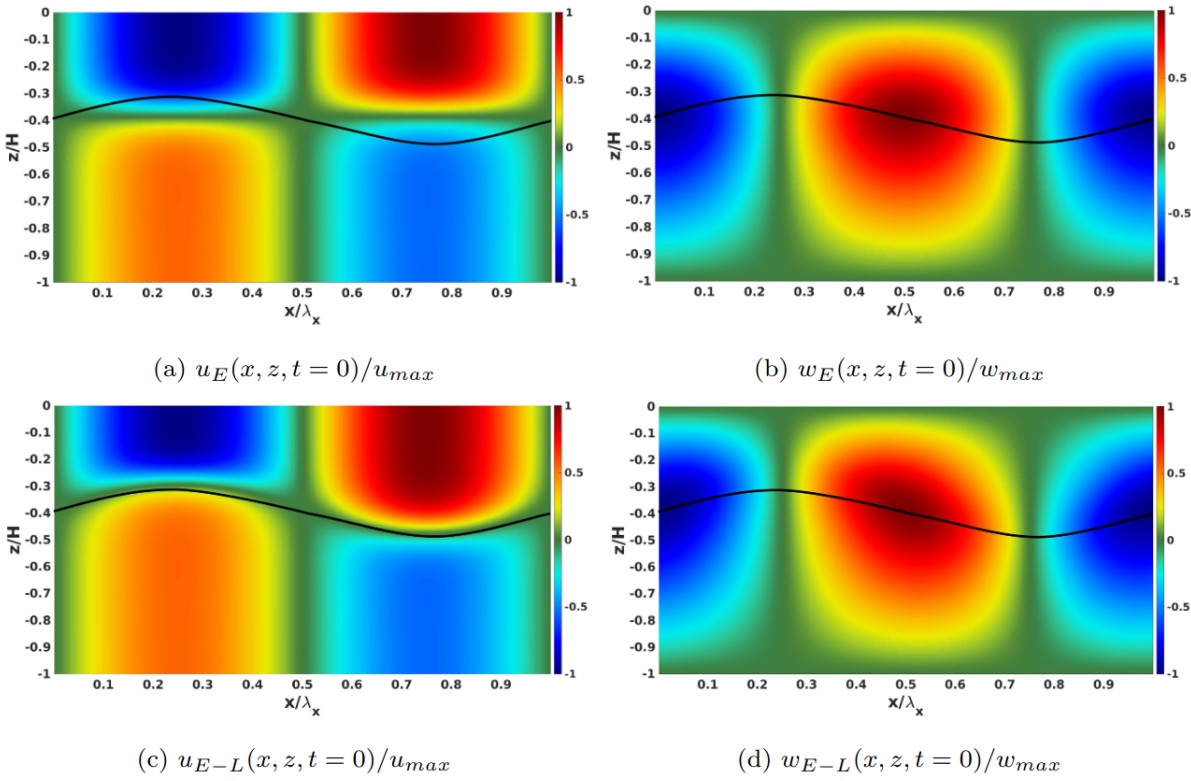

(a) $u_E(x,z,t=0)/u_{max}$     (b) $w_E(x,z,t=0)/w_{max}$

(c) $u_{E-L}(x,z,t=0)/u_{max}$     (d) $w_{E-L}(x,z,t=0)/w_{max}$

**Figure 3.** Snapshots of the velocity fields over one wavelength in the Eulerian approach (panels (a) and (b)) and in the Euler-Lagrange approach (panels (c) and (d)) at an arbitrary time $t = 0$ to highlight the spatial wave form. The displaced pycnocline is represented as a black line. Velocities are normalized with their maximum values.

**3.3 Euler-Lagrange approach**

When the Eulerian approach in Section 3.2 is applied to situations with a sharp pycnocline, a consistent error is observed for all but the smallest and shortest waves (see Section 3.4.2). This disintegration of the generated wave is due to the fact that the



up-and-down motion of the pycnocline is not accounted for in the vertical mode description. A natural way to account for this motion is achieved by introducing the wave induced displacement of the pycnocline. This is measured by the vertical displace-

ment $\eta(x, z, t)$ of the isopycnals (Fig. 2), or isopycnal displacement. Such an approach is labelled as *partially Lagrangian* since it follows the *vertical* displacement of individual fluid parcels through time. When combined with the usual description in the horizontal, it is labelled as an Euler-Lagrange approach (Gear and Grimshaw, 1983; Turkington et al., 1991).

Introducing the vertical displacement $\eta(x, z, t)$ of the isopycnals allows for a different, more natural description, of the perturbed stratification. Specifically, neglecting the uniform background density $\rho_0$ for the sake of compactness, the wave-

induced density field can be alternatively expressed as,

$$\rho(\boldsymbol{x}, t) = \bar{\rho}(z - \eta(x, z, t)), \tag{20}$$

which signifies that the density at a point is the same as that at an appropriate height far upstream. The density perturbation $\rho'_{E-L}$ can also be rewritten as,

$$\rho'_{E-L} = \bar{\rho}(z - \eta(x, z, t)) - \bar{\rho}(z). \tag{21}$$

Taylor expanding the right hand side shows that $\rho'_{E-L}$ is a polynomial in $\eta$ with the classical Eulerian description giving only the first term:

$$\rho'_{E-L} \approx -\eta \frac{d\bar{\rho}}{dz} + \frac{1}{2} \frac{d^2\bar{\rho}}{dz^2} \eta^2. \tag{22}$$

By effectively keeping more terms in the expansion, the efficacy of the deep water forcing is improved significantly. The first term carries a structure representative of the Eulerian approach in Eq. (19). However, from the second derivative of the

background density profile in the second term, we can see that Euler-Lagrange effects can be expected to be important for finite-amplitude waves when the stratification exhibits a sharp pycnocline. Such is exactly the case for the study at hand and the continuous two-layer stratification it uses (see Fig. 1 and Section 3.4.1).

At this juncture, it is worth emphasizing that the inclusion of the second term in the Taylor expansion of Eq. (22) still only provides an approximate solution of the governing equations (both the linear and nonlinear Euler equations) for the input wave

field. The purpose of this study is not provide an exact solution in this context, particularly in a nonlinear sense as enabled, e.g., by the internal-solitary-wave-generating algorithm of Turkington et al. (1991) as implemented in Dunphy et al. (2011). Instead, as will be subsequently be demonstrated, we are aiming for an approximate solution of the linear Euler equations that will drive the deep-water boundary forcing of finite-amplitude waves in a fully nonlinear simulation such that the waves can remain robust both near the source and further along the propagation path when nonlinear effects modify their waveform.

For the quasi two-layer continuous stratification case, an order of magnitude comparison between the two terms in the right-hand-side of Eq. (22) may be obtained if one uses as characteristic density and length scales the density jump, $\Delta\rho$, across the





pycnocline and the pycnocline thickness, $\delta$. Per Eqs. (26) and (29), as outlined in the next two sections, one may further write out the isopycnal displacement function as:

$$\eta(x,z,t) = A\frac{k}{\omega}r(x,z,t) \,, \tag{23}$$

where $A$ is the previously introduced amplitude factor and $r(x,z,t)$ is a structure function, harmonic in $x$ and $t$ and determined by the eigenfuction $W(z)$ in the vertical, which assumes values in the range $[-1,1]$. Using the characteristic scales above and Eq. (23), one can show that the ratio of the magnitude of the second term to that of the first one in the right-hand-side of Eq. (22) scales as $(A/\delta)(k/\omega)r(x,z,t)$. Therefore, the strength of the Euler-Lagrange effects becomes important when the parameter $(A/\delta)/(k/\omega) = A/(\delta c)$ is sufficiently larger than unity ; here $c = \omega/k$ is the linear phase speed obtained by solv-
ing the eigenvalue problem outlined in Sec. 3.1. The particular condition is satisfied for high wave amplitude, small pycnocline thickness and slow wave propagation speeds.

Restricting the scope to vertical mode-1 waves if $A/(\delta c)$, is sufficiently larger than unity, Euler-Lagrange effects also become important when the structure function $r(x,z,t)$ is $O(1)$ over a long enough horizontal length-scale. For very long waves, $\lambda/H >> 1$, this is the case over effectively the entire wavelength and $r(x,z,t) = r(z,t)$. For a wave with finite horizontal
wavelength, $\lambda$, which is a finite multiple of the water-depth ($H$), the along-wave variation of $r(x,z,t)$ needs to be retained. These two properties of the horizontal structure of $r(x,z,t)$ are at the crux of the formulations outlined in the next two sections.

Inserting the density perturbation Eq. (21) into Eq. (15), a definition analogous to the free surface kinematic boundary condition (Hodges and Street, 1999) arises for the isopycnal displacement $\eta$:

$$\frac{D\eta}{Dt} = w. \tag{24}$$

When deriving the actual velocity field using the Euler-Lagrange approach, the stratified waveguide analogy is still valid, but with a time dependent stratification to account for the wave-induced changes to account for the dynamic nature of the pycnocline. The derivation is the same as previously, only replacing the fixed frame of reference by the one tracking the displaced pycnocline. The decomposition presented in Eq. (10) is still valid, as the change of frame of reference considered only involves a vertical translation in $W$. Therefore the vertical velocity $w_{E-L}$ is now actually given by

$$w_{E-L}(x,z,t) = w_{Eul}(x,z-\eta,t) = -Ak\cos(kx-\omega t)W(z-\eta). \tag{25}$$

Note that in Eqs. (21) and (25) $\eta$ is defined using Eq. (26) or (29), as elaborated in Secs. 3.3.1 and 3.3.2.

### 3.3.1    Long waves

As a first approximation, since the waves of interest are long ($\lambda/H \gg 1$) and as outlined in the previous section, the $x$ dependence of the vertical displacement $\eta$ is neglected and $\eta$ *is considered to depend only on the vertical position $z$ and time $t$*. The





displacement of the fluid in the vertical direction along the wall $\eta(x = 0, z, t)$ is therefore computed by integrating in time the $w$ component of the velocity field. The Eulerian approach discussed previously then gives a good approximation, as a starting point, of the $w$ field (Eq. (17)), and $\eta$ may be derived as:

$$\eta(z, t) = \int_0^t w_E(x = 0, z, t) \, \mathrm{d}t = \frac{Ak}{\omega} \sin(-\omega t) W(z). \tag{26}$$

Using the continuity equation, Eq. (16), $u_{E-L}$ is derived accordingly. Specific attention needs to be paid to the $z$ variation
of the pycnocline's displacement $\eta$. In this regard, using the chain rule in differentiating $W(z - \eta)$, one obtains:

$$\frac{\partial u_{E-L}}{\partial x} = -\frac{\partial w_{E-L}}{\partial z} = Ak \cos(kx - \omega t) \left(1 - \frac{\partial \eta}{\partial z}\right) W'(z - \eta), \tag{27}$$

leading to,

$$u_{E-L} = A \sin(kx - \omega t) \left(1 - \frac{\partial \eta}{\partial z}\right) W'(z - \eta). \tag{28}$$

An extra term depending on the $z$-dependence of $\eta$ now appears in the expression of $u_{E-L}$ to account for the movement of
the pycnocline as contrasted to Eq. (18). Note also that the prime denotes a derivative with respect to the argument of $W$.

By comparing the spatial structure of the two approaches, (Fig. 3), the main difference resides in the structure of the velocity fields: in the Euler-Lagrange approach, the velocity field's deformation tracks that of the pycnocline's (plotted in black in Fig. 3 (c) and (d)), in contrast to the purely Eulerian case where the velocity field treats the pycnocline's position as constant (Fig. 3 (a) and (b)).

### 3.3.2   Waves of finite wavelength

So far, the dependence of $\eta$ on the along wave position $x$ has been neglected in a first approximation since the considered waves are assumed to be long. Practically, numerical simulations are often required to consider waves of finite wavelength. In this case, to accurately satisfy the continuity equation, Eq. (16), the $x$ dependence must be accounted for and the displacement $\eta$ is expressed as:

$$\eta(x, z, t) = \int_0^t w_{Eul}(x, z, t) \, \mathrm{d}t = \frac{Ak}{\omega} \sin(kx - \omega t) W(z). \tag{29}$$

Integrating Eq. (27) now needs to take into account the $x$ dependence of $\eta$, leading to a new expression for the horizontal velocity perturbation:

$$u_{E-L} = -\frac{\omega}{kW(z)} \left(1 - \frac{\partial \eta}{\partial z}\right) W(z - \eta) + \frac{\omega W'(z)}{kW^2(z)} \Phi(z - \eta) + B, \tag{30}$$





where $\Phi$ is the antiderivative of $W$ and $B$ is an integration constant. The second term in the right hand side of Eq. (30) appears to be, by evaluation of terms across the height of the domain, orders of magnitude smaller than the first one for a typically used wave and is therefore dropped:

$$u_{E-L} = -\frac{\omega}{kW(z)}\left(1-\frac{\partial\eta}{\partial z}\right)W(z-\eta)+B,$$
(31)

$B$ can be derived from the fact that the horizontal velocity is zero at the depth of the pycnocline, leading to:

$$B = \frac{\omega}{kW(z_p)}\left(1-\frac{\partial\eta}{\partial z}(0,z_p,t=0)\right)W(z_p-\eta(0,z_p,t=0)).$$
(32)

Since the right hand side of Eq. (31) is not defined on the boundaries for $z=0$ and $z=-H$, it can be extended using a continuous linear extension, finally resulting into:

$$u_{E-L}(z,t) = \begin{cases} \frac{\omega}{kW(z)}\left(1-\frac{\partial\eta}{\partial z}\right)W(z-\eta)+B & \text{for } 0 > z > -H \\ \frac{\omega}{k}\left(1-\frac{\partial\eta}{\partial z}(0,0,t)\right)+B & \text{for } z=0 \\ \frac{\omega}{k}\left(1-\frac{\partial\eta}{\partial z}(0,-H,t)\right)+B & \text{for } z=-H \end{cases}.$$
(33)

This approach will hereafter be referred to as *optimized Euler-Lagrange*.

Note that, when used to implement deep-water wave-generating boundary conditions, in all final expressions for $u_{E-L}$, $w_{E-L}$ and $\rho'_{E-L}$ derived in this section or Section 3.3.1, the value of $x$ is set to zero without any loss of generality, in a manner similar to the Eulerian approach as discussed in Section 3.2. In a similar vein, the associated adjustments introduced in the pressure boundary condition due to the presence of a time-dependent boundary-normal velocity field at the deep water boundary are discussed in Appendix B.

Finally, per the previous discussion of Eq. (22), we emphasize that, by construction, neither of the variants of the Euler-Lagrange approach outlined are designed as exact solutions to the linearized Euler equations: perfectly shaped monochromatic waves should not be expected. As illustrated by the results in Section 3.4.3, the waves generated through the most suitable of the two Euler-Lagrange approaches (dictated by the aspect ratio $\lambda/H$ at hand) are far more robust than those produced by the purely Eulerian approach. As a result, higher-amplitude longer waves can be used as forcing of fully nonlinear simulations (see Section 3.4.4).

## 3.4 Simulations of periodic internal waves in uniform-depth water

### 3.4.1 Numerical setup

Across all numerical simulations conducted in this study, the wave-based Reynolds number is held constant at $Re_w = 2.5\times10^5$. Such a value of $Re_w$ is representative of laboratory scale, yet is sufficiently high to avoid any attenuation in wave amplitude





in the propagation zone. The Schmidt number $Sc = \nu/\kappa$ is fixed at unity. A wave-based Froude number of $Fr = 0.2$ is linked
to generated waves that may confidently be characterized as finite-amplitude and can support the development of sufficiently
strong nonlinear effects as they propagate away from the forcing boundary.

The quasi two-layer continuous stratification profile for $\bar{\rho}(z)$, given by Eq. (3), is kept the same across all runs. A relatively
thin pycnocline with $\delta/H = 0.09$ is used with a non-dimensional density jump across the pycnocline of $\Delta\rho/\rho_0 = 2N_0^2\delta/g = 1.7 \times 10^{-3}$ located at a relative position $z_p/H = -0.4$. The particular value of $\delta/H$ is chosen to mimic the thinner pycnocline
of the early Fall stratification profile in a long-deep lake (Schweitzer, 2010). The deep-water-generated wave used in these
simulations is chosen to have an aspect ratio $\lambda_x/H = 10.12$. Such a value of $\lambda_x/H$ qualifies the wave as finite-length, albeit
not short. Finally, for the particular thin-pycnocline stratification profile and choice of $\lambda_x$, the amplitude coefficient $A$ leading
to a value of $Fr = 0.2$ corresponds to a value of $A/(\delta\ c)\ =\ 5$. Euler-Lagrange effects will clearly be present. The choice of
$\lambda_x/H$ further motivates the question as to whether the fully optimized Euler-Lagrange approach is needed.

All simulations are performed in a uniform-depth tank of depth $H$ and length $L = 10\lambda_x$. The domain is chosen sufficiently
long to allow the development of nonlinear effects within the generated waves. Using uniformly sized rectangular spectral
elements, 224 points per wavelength $\lambda_x$, are employed in the horizontal direction, whereas 161 points span the entire water
column in the vertical direction. The resolutions are given in Table 1 and the elements are uniformly spaced in both length
and height. The internal grid point distribution in each element consist of non-uniformly distributed two-dimensional Gauss-
Lobatto-Legendre (GLL) integration points (Canuto et al., 2007).

| Polynomial order $p$ | 7 |
|---|---|
| Number of elements in the $x$-direction $m_x$ | 120 |
| Number of elements in the $z$-direction $m_z$ | 20 |
| Total number of points in the $x$-direction $N_x$ | 2241 |
| Total number of points in the $z$-direction $N_z$ | 161 |
| $\Delta x/H$ range | $[0.0181 - 0.0656]$ |
| $\Delta z/H$ range | $[0.0025 - 0.0091]$ |

**Table 1.** Grid point count and resolution for the two-dimensional simulations in uniform depth tank.

### 3.4.2 Limitations of the Eulerian approach

The limitations of the Eulerian approach for the wave forcing are visible in a linear INSE solver (not shown here) but are more
readily demonstrated in the framework of nonlinear solver of this type. As shown in the left panel of Fig. 4, the Eulerian-
generated $u$-velocity field, namely the shear-layer between the upper and lower lobes of opposite velocity, tracks horizontally
along the location of the undisturbed pycnocline (similar to the top left panel of Fig. 3) and does not follow the actual pycno-
cline location (see bottom left panel of Fig. 3). Immediately visible non-physical numerical features emerge near the forcing
boundary at near-grid-scale, as evidenced by the lobes of alternating sign in the vertical velocity in that region. These spurious



vertical velocities are a factor of two larger than the theoretically prescribed ones within the target wave leading to a commensurate reduction of the time-step by virtue of the CFL condition. Finally, non-negligible regions with density values that

exceed by a factor of 2-to-2.5 the bounds of the background stratification are observed (see the blanked-out regions in the right Fig. 4 right panel). These spurious numerical effects intensify as more waves are generated for this value of $Fr = 0.2$. Further intensification of these effects is observed at $Fr = 0.5$ (not shown). In this case, the non-physical nonlinear interactions are strong enough to further amplify the near-source spurious vertical velocities and cause an aggressive, and prohibitive, reduction of the timestep .

Numerical experimentation indicates that for cases with a well–defined pycnocline the Eulerian approach produces robust waves, and is effectively only valid, for small amplitude ($Fr \lesssim 0.05$, $\eta_{max}/H \lesssim 0.02$) and short wavelength waves ($\lambda/H \lesssim 5$). It is important to note that the ad hoc linearizing that leads to the Eulerian approach is effectively a linear truncation of a Taylor series. In the amplitude tending to zero limit the Eulerian and E-L approaches match. However, even at moderate amplitudes a significant mismatch is observed (see black dotted lines in Fig. 4). Our approach retains the notation of the Eulerian approach,

which is easier to implement in a software setting, but effectively implements the higher order terms in the Taylor series (at the cost of some algebra).

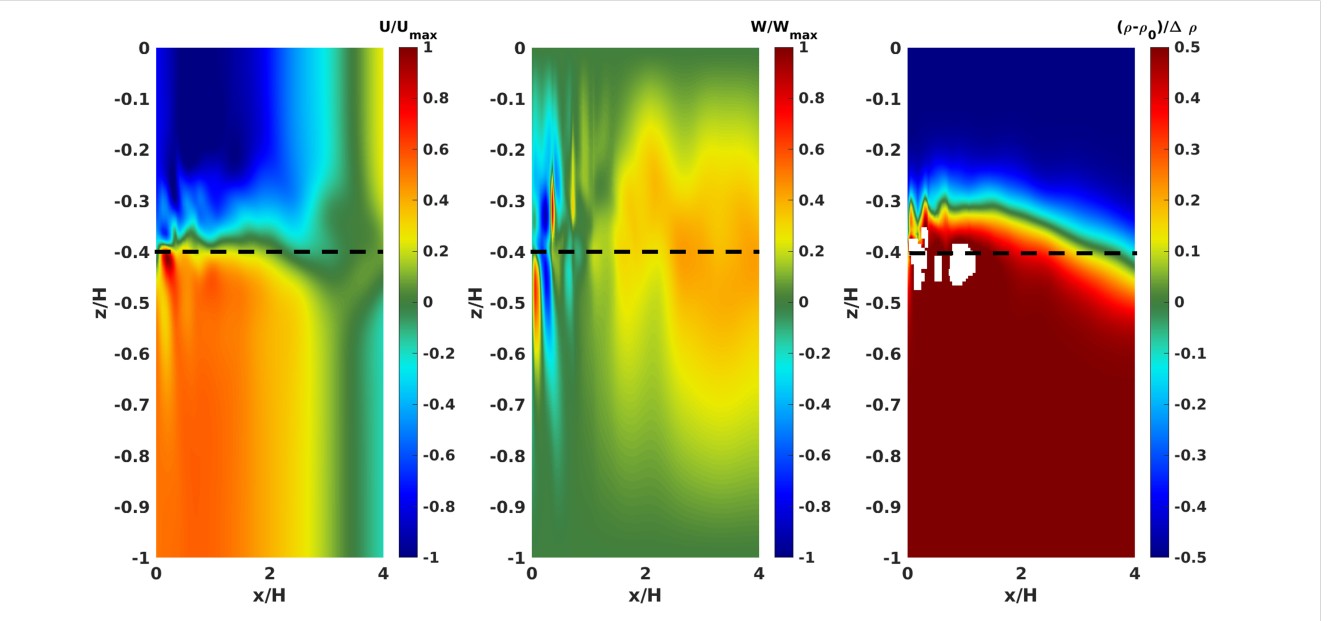

**Figure 4.** Fully non-linear simulation using Eulerian wave generation boundary conditions at $t/T = 5.8$ for $Fr = 0.2$ and $\lambda/H = 10$. The initial undisturbed pycnocline location is represented as a black line. Velocities are normalized with their maximum values and the adjusted density by the density jump at the pycnocline. Near-source near-grid-scale lobes of vertical velocity are a factor of 2-2.5 larger than that of the prescribed wave. White regions in the density contours are correspond to values exceeding the colorbar limits which are set by the undisturbed background density profile.




### 3.4.3 Linearized Navier-Stokes simulations

To differentiate the features strictly resulting from the differences in the wave generation approach from the ones resulting from the nonlinear effects downstream of the source, the first set of simulations are restricted to solving the *linearized* incompressible
Navier-Stokes equations, under the Boussinesq approximation. The nonlinear terms have been dropped in Eqs. (4), (5) and (6) analogously to what has been done in the linearized Euler equations, i.e., Eqs. (14), (15) and (16). Per the discussion of the previous section on the limitations of the Euler approach, only the Euler-Lagrange approach is considered here.

The sensitivity to including the $x$-dependence of the isopycnal displacement $\eta$ in the Euler-Lagrange approach is assessed in Fig. 5 by examining the velocity and density fields that are produced by the approaches outlined in Sections 3.3.1 and 3.3.2 per
the corresponding expressions for $u_{E-L}$ in Eqs. (28) and (33). The $w_{E-L}$ and $\rho'_{E-L}$ forcing functions have the same structure in both cases and are given by Eq. (25) and Eq. (21), noting any adjustments for the $x$-dependence of $\eta$ in the optimized Euler-Lagrange formulation. Results are shown after approximately 7 wave periods since the initiation of deep-water boundary wave forcing.

Close to the wave source, periodic shorter-wavelength features are observed for both approaches. These smaller-scale os-
cillations result from neither of the Euler-Lagrange approaches being an exact solution of the linearized Euler equations, as discussed in Sec. 3.3. The amplitude and downstream persistence, however, of these shorter-wavelength effects is markedly weaker in the optimized Euler-Lagrange approach right panels in Fig. 5, because of the finite wavelength of the generated wave (see Sec. 3.3). To this end, the optimized Euler-Lagrange approach is the method of choice in the fully nonlinear simulations, given that our baseline wave has an aspect ration of $\lambda/H$: using it minimizes any possible non-linear interactions between the
above parasitic smaller-scale waves and the main target wave which otherwise pose non-trivial challenges for the robustness of the latter wave.

### 3.4.4 Fully nonlinear simulations

The resulting velocity and density fields, obtained by solving the fully nonlinear Navier Stokes Equations under the Boussinesq approximation (Eqs. (4), (5) and (6)), forced by the optimized Euler-Lagrange approach are shown in Figs. 6 after 10 wave
periods. The spurious numerical features close to the boundary have been found to be significantly weaker (not shown here) as compared to what is shown in Fig. 4. Whereas the shear-layer of the horizontal velocity field tracks the oscillating pycnocline according to Fig. 3, and doesn't affect the wave generation. Additionally, no spurious mass generation is observed, with density values restricted within the limits dictated by the background stratification. Finally, near-grid-scale vertical velocity near the source remain very small in magnitude such that the wave-induced vertical velocity is the only factor controlling the
timestep, as expected. The nonlinear response of the generated wave may now be examined along the propagation path without contamination by spurious nonlinear interactions due to small-scale near-source transients.

The structure of the generated waves is indeed visibly modified by nonlinearity as they propagate away from their source, with different waveform geometries becoming immediately identifiable as a function of distance from the source. Fig. 7 attempts to offer such a waveform classification across three different sub-windows along the propagation path. Fig. 7(a) shows



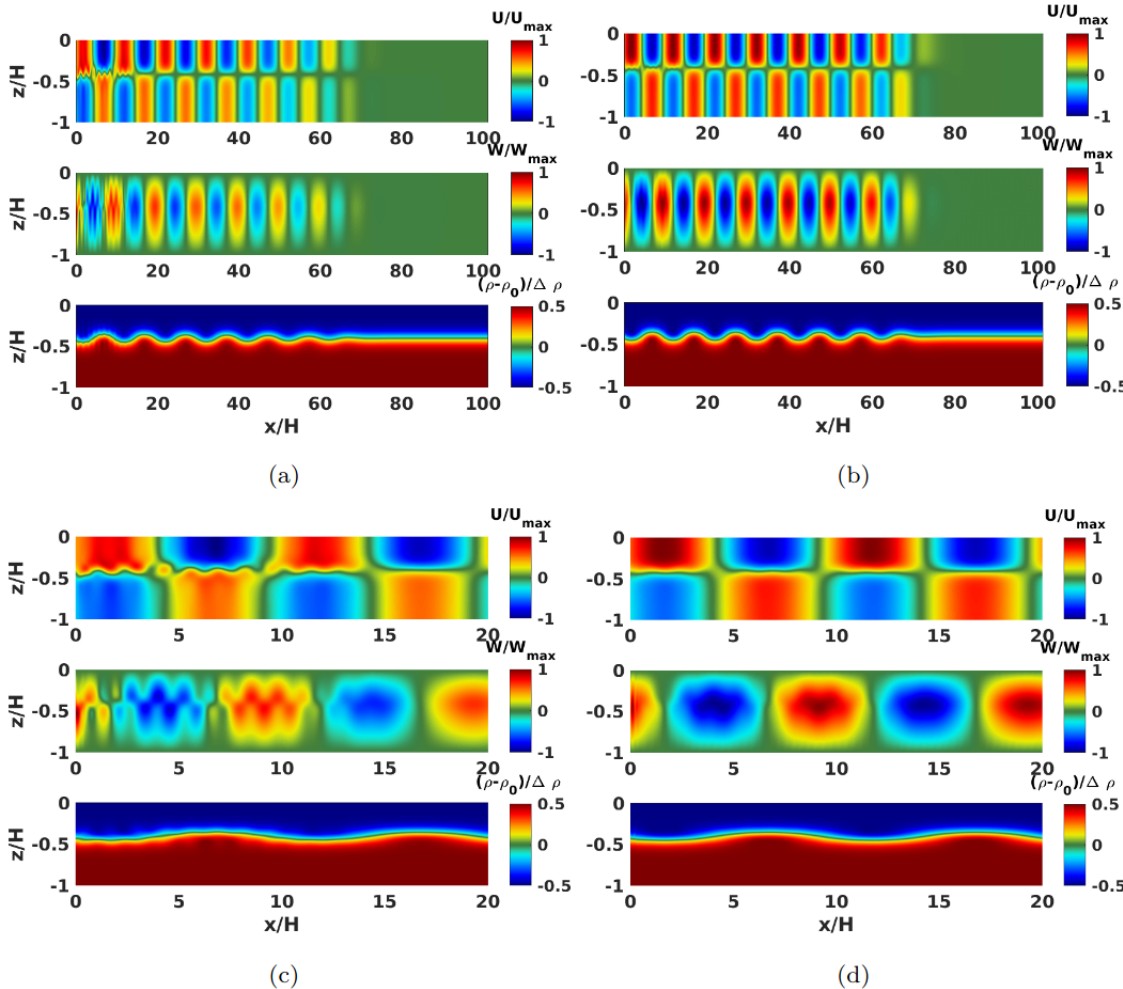

**Figure 5.** Comparison of the velocity and density structure generated by the two different Euler-Lagrange wave generation approaches in a fully linear simulation at $t/T = 7.4$. The left panels (a) and (c) use the Euler-Lagrange approach and the right panels (b) and (d) use the optimized one. The bottom panels (c) and (d) correspond to a zoomed view of the respective top panels (a) and (b). Velocities are normalized with their maximum values and the adjusted density by the density jump at the pycnocline.





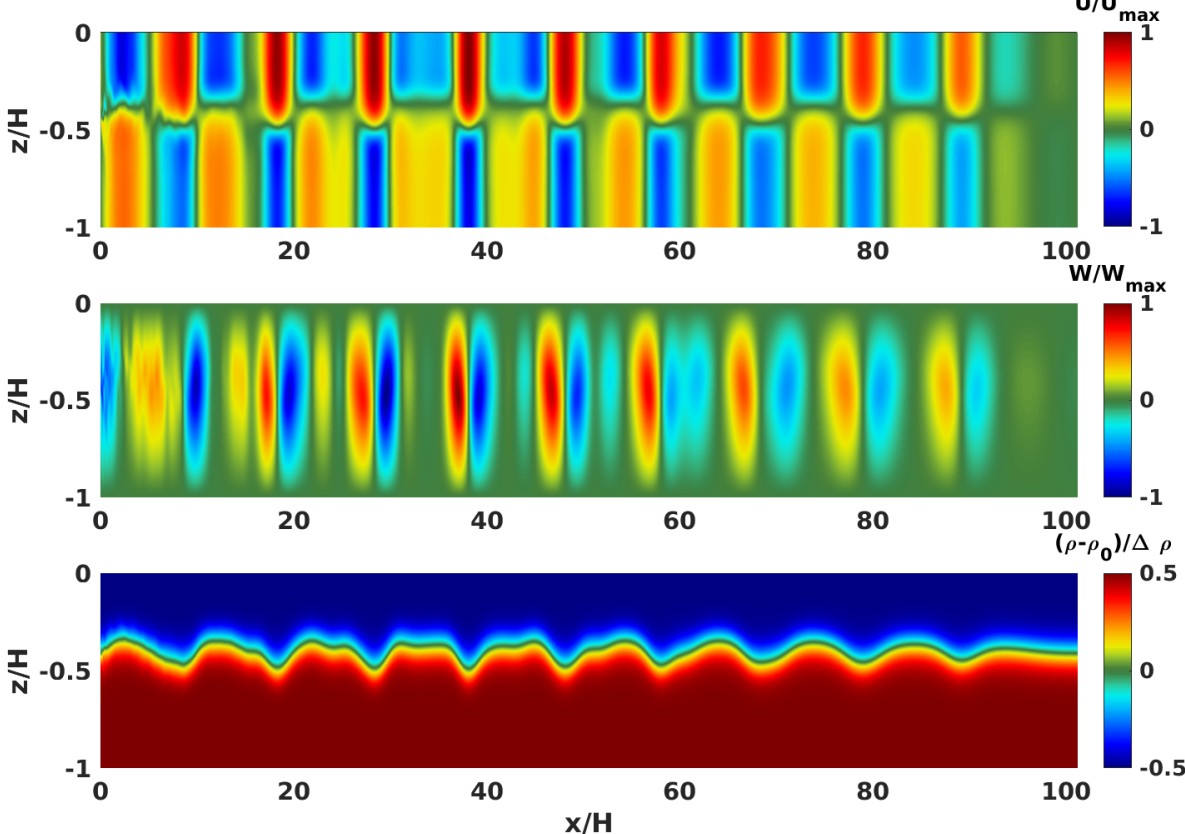

**Figure 6.** Fully non-linear simulation using optimized Euler-Lagrange wave generation boundary conditions at $t/T = 10$. Velocities are normalized with their maximum values and the adjusted density by the density jump at the pycnocline.

waves of depression that develop close to the source. Since the particular waves have large flat plateaus and narrow troughs, a clear similarity with cnoidal waves (Boyd, 2015) is suggested. During a transitional phase, shown in Fig. 7(c), the wave troughs broaden. Further downstream the waves tend to assume a near-sinusoidal shape with peaks and troughs of comparable width (Fig. 7(e) ).

A more quantitative description of the different types of observed waveforms is enabled by examining the corresponding

along-wave spectral content. One-dimensional spatial fast Fourier transforms (FFT) of the density are computed for each sub window, focused at the depth of the undisturbed pycnocline, which in one-dimensional streamwise Fourier spectra of the density $\hat{\rho}(k)$. Special attention needs to be paid when computing the FFTs for the simulations at hand, since the internal grid point distribution within each spectral is non-uniform (Section 3.4.1). A non-uniform FFT algorithm is therefore used (Dutt and Rokhlin, 1993; Potter et al., 2017), as it is well tested and readily available.

Closer examination of the right column of panels in Fig. 7, suggests that the along-wave spectral content has power spectral density in regions not specified by the forcing. In particular, a strong second harmonic persists at a downstream distance as





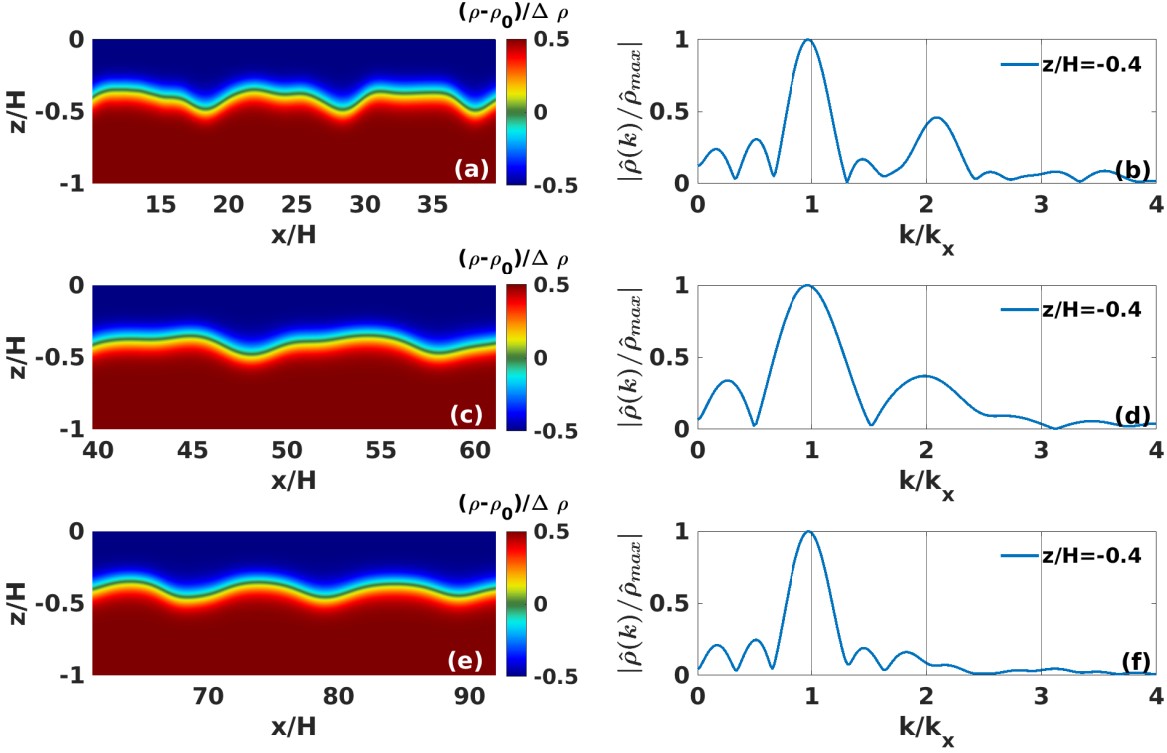

**Figure 7.** Exploded view of full density contours at different downstream locations for the waves shown in Fig. 6 illustrating the development of wave forms (panels (a), (c) and (e)). Respective streamwise Fourier spectra $\hat{\rho}$ of $\rho(x)$ computed at $z/H = z_p = -0.4$ and $t/T = 10$ normalized by the maximum peak depending on the wavenumber normalized by the prescribed wave number $k_x$ (panels (b), (d) and (f)).

large as $60H$. Further downstream from the forcing boundary, the amplitude of this harmonic significantly attenuates resulting in a wave that is closer to being monochromatic (as confirmed by the visualization of Fig. 7 (e)). In the context of a fully non-linear simulation with a sloping boundary, adjusting the length of the section of the computational domain over which the waves propagate prior to reaching the slope allows one to decide how much waves are allowed to naturally adjust due to their finite amplitude and dispersion. Equivalent simulations, which separate slope from source by only a fraction of the horizontal wavelength (Masunaga et al., 2015, 2016), are not expected to support a nonlinearly adjusted (and potentially steepened) waveform as the incident wave reaches a slope.

The vertically integrated kinetic energy ($KE$), at any down-stream position $x$, is:

$$KE = \int\limits_{-H}^{0} \frac{1}{2}\rho_0(u^2 + w^2)dz. \tag{34}$$




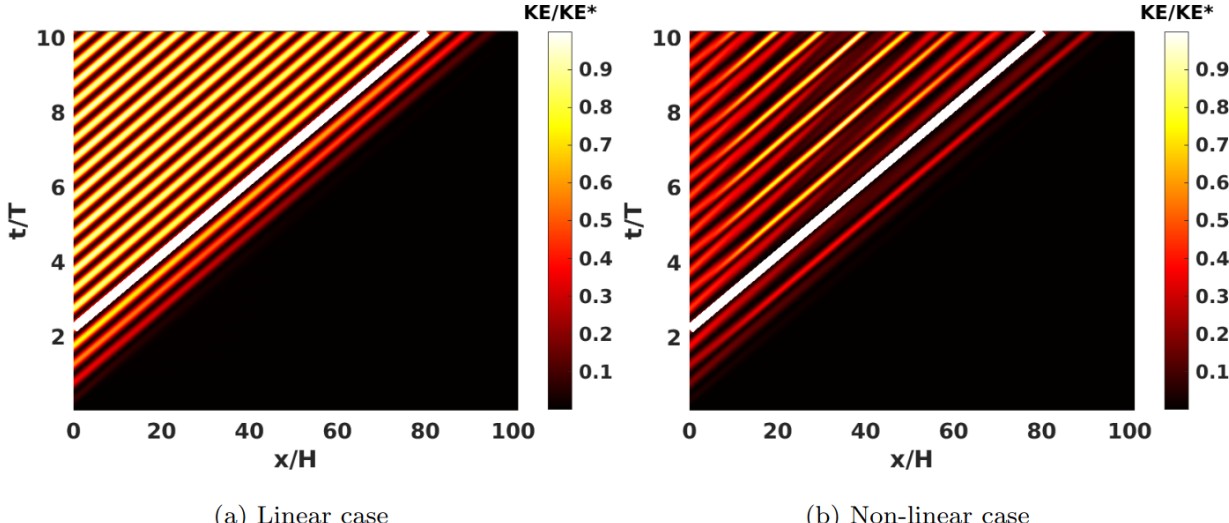

(a) Linear case  (b) Non-linear case

**Figure 8.** Vertically integrated kinetic energy $KE$ normalized by its maximum value $KE^*$ as a function of the downstream position and time for both the linear and non-linear case. The white line corresponds the theoretical energy transport characteristic $x = c \times t$.

$KE$ is shown as a function of time and position for both the fully linear case Fig. 8 (a) and non-linear case Fig. 8 (b). The theoretically prescribed characteristic of energy transport given by the wave speed $c = \lambda_x/T$ is also plotted. Fig. 9 presents the interpolation of the kinetic along the prescribed characteristic of energy transport in the linear case shown in white in Fig. 8 (a). The slope of the $KE$ contours, a measure of the group velocity, appears to match well with the theoretical value. Viscous decay can be considered negligible since the wave-based Reynolds number is chosen $Re_w = (\lambda_x^2/\nu)/T \gg 1$. The deviation along the characteristic in Fig. 9 can therefore be attributed to the dispersive aspect of the continuous two-layer stratification. In the non-linear case, characteristics also appear to be parallel to the theoretical solution, even if the $KE$ is not constant along it due to non-linearities generating extra wavelengths.

## 4  Discussion

The generation process of finite-amplitude periodic waves through time-dependent deep-water Dirichlet boundary condition has been examined for the case of a quasi two-layer continuous stratification (Fig. 1). In the case of a linear stratification, $N^2$ is constant in time and space and therefore Eq. (11) has an analytical solution $W_n$. Such a solution leads to an exact expression of the perturbation fields for the linear Euler equations under the Boussinesq approximation with an analytical vertical structure, with an explicit time dependence (see Appendix A). The amplitude of the generated waves can non-trivially exceed the limit prescribed by linear theory without any impact on wave robustness as evidenced by the deep-water waves used in the linearly stratified simulations of Winters (Winters, 2015).



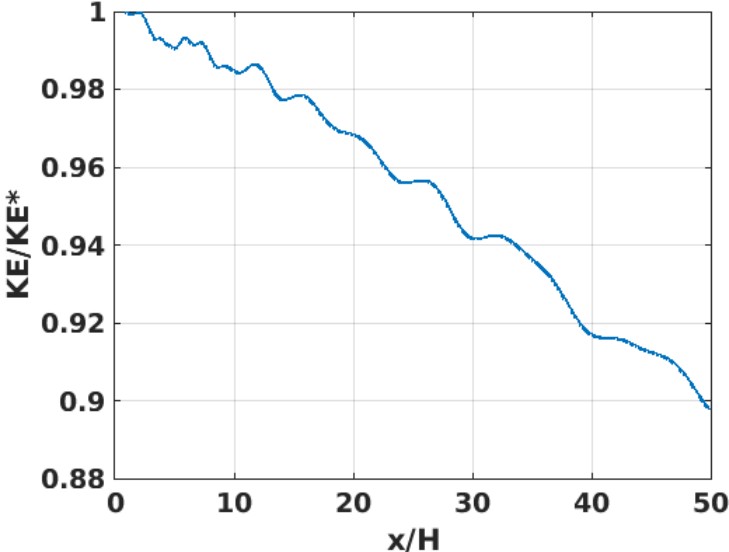

**Figure 9.** Kinetic energy along the prescribed characteristic of energy transport in the linear case shown in Fig. 8 (a).

As described in Section 3, the quasi-two-layer stratification studied here appears to be more complex. In this context we don't have an analytical expression for $W(z,t)$, leading to the different approximations introduced in this study. These extra layers of approximations will therefore tighten the amplitude limitations of the wave that can be generated. Nevertheless depending on the chosen stratification, numerical experiments (not shown here) have demonstrated that robust waves up to $Fr = 0.5$ are achievable using the optimized Euler-Lagrange approach.

Another important feature that has been demonstrated by Fig. 7, is the fact that in the quasi two-layer continuous stratification the forcing produces wave-trains that are non-monochromatic. Eq. (33) reveals that the optimized Euler-Lagrange approach results from the multiplication of temporary-oscillating terms, leading to the appearance of multiple harmonic wavelengths in the generated wave. Sufficiently downstream of the source, the strength of these harmonics seems to diminish for the $Fr = 0.2$ waves shown in Fig. 7(v), with wave-induced perturbations that may be regarded as assuming a near-sinusoidal waveform. Nonetheless, the wave-trains at the same downstream location in our experiments with $Fr = 0.5$ (not shown here) are found to remain remarkably nonlinear with extremely steep fronts therein.

Per the literature review in the introduction, the only other computational study considering the generation of long finite-amplitude waves in a two-layer stratification, with sufficient distance for the waves to develop downstream from the source, that the authors are aware is that of Dauhajre et al. (2021). We suspect that no issues were reported in regards to the deep-water generated waves for two reasons. First, motivated by apparently different objectives than this study, the use of wave aspect ratio of $\lambda/H = 200$ is remarkably long and, most likely, restrictive if a turbulence-resolving capability (and not a turbulence parameterization) is preferred. Additionally, the inferred normalized pycnocline thickness value of $\delta/H = 0.35$ is more representative of that in the oceanic continental shelf and not of a deep and long seasonally stratified lake, the primary motivator of this study.



Most importantly, noting that $Ak = W_{max}$, where $W_{max}$ is the maximum wave-induced vertical velocity, Dauhajre et al. work with a typical value of $A/(\delta c) \approx 1.8$. These non-dimensional parameter values along with any reduction of curvature at the base of the pycnocline through the insertion of a weakly, yet non-trivially, stratified lower layer may diminish the intensity of any Euler-Lagrange effects per Eq. (22) and the associated discussion. Finally, it is unclear how the numerical dissipation
built into the K-Profile-Parameterization (Large et al., 1994) actively used by Dauhajre et al. (2021) may have damped out any near-source short-wavelength initialization transients such as those reported in Section 3.4.3.

## 5   Conclusions

This study has examined the formulation of robust finite amplitude periodic internal-wave generating boundary conditions for a nonlinear stratification, highlighting extra levels of subtlety compared to the linear stratification case, while relying on
a higher-order accuracy spectral element method to discretize the governing equations. The commonly used Eulerian approach, which relies on a fixed reference frame, is found to develop non-trivial errors when implemented in simulations with a sharp quasi-two-layer continuous stratification and higher-amplitude internal waves with a horizontal-wavelength-to-depth (wave aspect) ratio that is finite, albeit not excessively large. The errors result because the prescribed wave forcing assumes a fixed/unperturbed pycnocline and does not account for the upstream and vertical wave-induced displacement of the pycnocline.
This mismatch between fixed wave-forcing and moving pycnocline is shown to scale with the parameter $A/(\delta c)$, where $A$ is a measure of wave amplitude, $\delta$ is the pycnocline thickness and $c$ is the wave propagation speed. Simulations with values of $A/(\delta c) = 5$ show spurious mass generation near the wave-generating source with accompanying unphysical near-grid-scale vertical velocities that can detrimentally reduce the computational timestep, and even prohibitively restrict it for a long enough time and higher values of wave-induced Froude number, $Fr$. The minimal numerical dissipation of the spectral element method
can not damp these spurious numerical features.

For values of $A/(\delta c)$ sufficiently larger than unity, an Euler-Lagrange approach needs to be used instead, which does account for the above pycnocline displacement, in the wave generation. Although an exact solution of the linearized Euler equations under the Boussinesq approximation is not actually attained through this approach, the resulting waves are sufficiently robust: they can propagate away from the source ; nonlinear adjustments of their waveform is possible through leveraging the higher-
order-accuracy spectral element scheme.

The findings of this study will serve as a platform to enable a detailed numerical study of internal swash zones (ISZ), zones driven by the interaction of long periodic nonlinear internal waves with a sloping boundary. Such simulations will aim to investigate the parameter space in two-dimensions which would include wave Froude number, pycnocline thickness and depth, wave-aspect-ratio, slope value and the role of no-slip vs. free-slip boundary conditions, particularly on the slope. Select two-
dimensional studies will operate as the springboard for full-scale three-dimensional turbulence-resolving simulations These larger simulations may invariably be restricted by existing computational resources to wave aspect ratios in the range [20, 40]. As our interests are motivated by internal swash zones in seasonally stratified deep lakes lakes, we will use a two-layer continuous stratification with thinner pycnoclines typical of such environments (Schweitzer, 2015). As such, the parameter



$A/(\delta c)$ will be non-trivially larger than unity. To address this region of parameter space, an Euler-Lagrange approach is needed
to account for the wave-induced displacement of the isopycnal field in both vertical and horizontal directions. The optimized
Euler-Lagrange approach will be used to generate robust high-amplitude deep-water internal waves, at values of Froude number
up to $Fr = 0.2$.

A parallel avenue of future investigation of the findings of this paper may be their translation to experimental internal
wave generators. Horizontally oscillating paddles are reported as limited to significantly short waves with aspect ratio around
10 (Ghassemi et al., 2022). The vertically-stacked-plate/eccentric-camshaft structure of Mercier, Gostiaux and co-workers
(Mercier et al., 2010; Gostiaux et al., 2006), through its ability to reproduce a baroclinic structure in the vertical, may be the
most amenable experimental technique to adopting aspects of the optimized Euler-Lagrange approach presented here.

**Appendix A: Details on the derivation of the vertical structure in the linear stratification case**

In the case of a linear stratification, we have by definition:

$$N(z) = N_0 \tag{A1}$$

Eq. (11) becomes a classic second order linear differential equation, analogous to that of a simple harmonic oscillator. The
solution is oscillatory in $z$:

$$W_n(z) = \sin(\frac{n\pi z}{H}) \ \ with \ k_n = \frac{n\pi}{H} \left( \frac{\omega^2}{N^2 - \omega^2} \right)^{1/2} \tag{A2}$$

**Appendix B: Subtleties of the implementation of the time-dependent deep-water boundary conditions**

To implement the different time-dependent deep-water boundary conditions approaches described in Sections 3.2 and 3.3, the
eigenfunction $W(z)$ and corresponding eigenvalue $k$ need to be calculated from Eqs. (11) and (12). A high-order spectral
element method (Diamantopoulos et al., 2022) is used to for this purpose.

The values of the eigenfunction, and its vertical derivative, on locations offset from the actual grid points, $W(z - \eta)$ and
$W'(z - \eta)$ are required in either of the Euler-Lagrange approaches (see Section 3.3). These values are obtained at each time
step through a cubic spline interpolation in the vertical.

To account for the wave-driven time-dependence of the deep-water vertical boundary, the corresponding pressure Neumann
boundary condition needs to be updated accordingly at each time step along the wave generating boundary, regardless of the
generation approach used. Per the projection of Eq. (4) onto the deep-water boundary, the new boundary condition for the
pressure results from the balance of the wall normal pressure gradient with all the other terms including, now in particular, the
temporal derivative of the velocity on the wave-generating boundary. The adaptation of this boundary condition for the pseudo-
pressure, the average pressure integrated over one time-step, in the context of the temporal discretization used (Karniadakis
et al., 1991) is straightforward.



Additionally, to reduce transient-driven contamination of the generated deep-water waves and force both velocity components and density perturbation to be zero at the deep-water boundary at time $t = 0$, the amplitude of the boundary forcing is
ramped up in time through application of an exponential envelope. The three forcing expressions for $u$, $w$ and $\rho'$ (see Sections 3.2 and 3.3) are multiplied by an envelope function, $f(t)$, defined by:

$$f(t) = \left( 1 - exp\left( \frac{t}{\tau} \right) \right) \qquad \text{(B1)}$$

where $\tau$ is a characteristic time scale of the ramp-up constrained by $\tau \ll T$ and set to $\tau = \frac{5}{100}T$ in this study

*Code and data availability.* All simulation output and data analysis scripts will become publicly available on a dedicated Online Filer Server,
under construction in collaboration with Cornell's Center for Advanced Computing, as mandated by the research grant Data Management Plan submitted to the U.S. National Science Foundation. The Online File Server is expected to be operational by early May 2024. Codes can be made available upon request.

*Author contributions.* **Pierre Lloret:** Conceptualization, Data curation, Investigation, Methodology, Software, Visualization, Writing - original draft. **Peter J. Diamessis**: Conceptualization, Funding acquisition, Project administration, Resources, Supervision, Writing - original
draft. **Marek Stastna**: Conceptualization, Writing - original draft. **Greg N. Thomsen**: Software, Writing - review & editing.

*Competing interests.* The authors declare that they have no conflict of interest.

*Acknowledgements.* Financial support is gratefully acknowledged from National Science Foundation - Division of Ocean Sciences (OCE) Grant 1948251. This work used Anvil2 at Rosen Center for Advanced Computing through allocation EES200010 from the Advanced Cyberinfrastructure Coordination Ecosystem: Services & Support (ACCESS) program, which is supported by National Science Foundation grants
#2138259, #2138286, #2138307, #2137603, and #2138296. Discussions on generation and breaking of internal waves in internal swash zones with Profs. Edwin (Todd) Cowen and Erika McPhee Shaw and Dr. Seth Schweitzer are gratefully acknowledged.



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
