# Peer review of "A robust numerical method for the generation and propagation of periodic finite-amplitude internal waves in natural waters using high-accuracy simulations"

_EGUsphere, 2024_

## Author Response (AR1)

**Response to Referee-1**

(Original comments are written in italic)

**Preface**

*In this paper, an Euler-Lagrange approach is developed and implemented to generate robust high-amplitude periodic deep-water internal waves. The near- and far-source robustness of the optimized Euler-Lagrange approach is demonstrated for finite amplitude waves in a sharp quasi two-layer continuous stratification representative of seasonally stratified lakes. The findings of this study will serve as a platform to enable a detailed numerical study of internal swash zones (ISZ), zones driven by the interaction of long periodic nonlinear internal waves with a sloping boundary. I recommend this paper for moderate revision before accepted for publication in NPG.*

We are thankful to the referee for having thoroughly read our manuscript. We really appreciate their positive feedback and their initial recommendation in favour of publication after moderate revisions. Hereafter, we will try to address the reviewer's insightful question and comments.

**Response to main questions**

1. *Why does the Euler method have a large error while the Euler-Lagrange method has a small error? What is the main reason?*

   From a mathematical standpoint, the main difference between the two approaches lies in the consideration of an extra term in the Taylor series expansion of the density perturbation, as already shown in Eq. (30), which also accounts for the nonlinearity of the background density profile. From a physical viewpoint, the Euler-Lagrange approach does provide for a wave-forcing that tracks the wave- induced pycnocline displacement unlike the Euler approach. The latter approach assumes a time-fixed density profile as shown in Eq. (20). To further emphasize this physics-based distinction between the two approaches, we have now provided an extra panel in Figure 3. This additional shows how the vertical eigenfunctions computed for the wave-displaced stratification at the wave peak and trough are offset from the corresponding eigenfunction computed for the initial undisturbed stratification.

More detail on the vertically offset eigenfunctions associated with the Euler-Lagrange approach along with the mathematical underpinnings may be found in the newly inserted reference by Lamb (1999).

2. *Can the established numerical simulation method be used to simulate in more complex stratification and with topography such as a slope? Is it possible to provide these preliminary simulation results?*

The established numerical simulation method has indeed used to simulate more complex stratification and topographies such as a slope. Two papers presenting results and analysis of two-dimensional and three-dimensional (turbulence-resolving) simulations of internal swash zones over multiple wave periods are in preparation. These papers constitute, in draft form, two chapters in the first author's PhD thesis submitted to Cornell University in late June. The thesis is expected to become available online through *e-Cornell* in the early Fall.

**Response to minor comments**

1. *Title "A robust numerical method for the generation and simulation of periodic finite-amplitude internal waves in natural waters" - the generation and propagation?*

Per the reviewer's recommendation, the title has been changed to "A robust numerical method for the generation and propagation of periodic finite-amplitude internal waves in natural waters using high-accuracy simulations".

2. *Line 181: The nodal spectral element method should be introduced in more detail;*

Per the reviewer's recommendation, more details on the spectral element method were introduced with the addition of an Appendix describing the treatment of the Poisson equation as well as the viscous term.

3. *Line 190: time dependent, vertically variable Dirichlet conditions, please provide detailed explanations;*

More details on the definition of the boundary conditions have been added in Part 2.2.

4. *Figure 4, Figure 6: Provide sub-figures (panels) numbers in Figure;*

Panels numbers have been added.

5. *Line 558-673: The references format needs to be further checked, and unified;*

The reference format has been unified.

**Response to Referee-2**

(Original comments are written in italic)

**Preface**

*The paper discusses the development and implementation of boundary conditions for simulating periodic finite-amplitude internal waves in a quasi two-layer continuous stratification using a spectral-element-method-based on incompressible flow solver. It highlights the limitations of the Eulerian approach, which can produce numerical artifacts in nonlinear stratifications, and introduces an Euler-Lagrange method that maintains wave integrity by accounting for isopycnal displacements. The study demonstrates the robustness of this approach in simulating internal waves and their interactions, providing a foundation for two-dimensional and, ultimately, three-dimensional turbulence-resolving simulations. I think the paper is interesting and can be published after minor changes.*

We are thankful to the referee for having thoroughly read our manuscript. We really appreciate their positive feedback and their initial recommendation in favour of publication after minor changes. Hereafter, we will try to address the reviewer's insightful comments.

**Response to the Referee's comments**

1. *The general numerical method description in section 2.4 needs more detail. Equations (4-6) contain a non-hydrostatic pressure gradient that requires an additional equation and corresponding boundary conditions for the pressure.*

   A more detailed description of the numerical method and, in particular, the boundary conditions for the Poisson pressure equation their adaptation to account for the time-dependent boundary wave forcing has been inserted in section 2.4.

   We admit that we are confused by the Reviewer's reference to a separate equation for a non-hydrostatic pressure gradient. As explained in the text, we solve for the perturbation, $p'$, to the background hydrostatic pressure which is in balance with the undisturbed background stratification. The balance between the latter two fields has been subtracted out of the momentum equations. The pressure perturbation, $p'$, itself involves a hydrostatic and non-hydrostatic component as explained by Smyth and Moum (J. Fluid Mech. 2006). Decomposing $p'$ into these two individual contributions is not a part of the numerical solution of the governing equations and is out of the scope of this paper.

2. *The generation of Dirichlet-type periodic boundary conditions is a key component of the paper. However, boundary conditions are only schematically indicated in Figure 2. Boundary conditions, as a main component of the study, should be presented in the governing equation section 2.2 and in the description of the Euler and Euler-Lagrange approaches in sections 3.2 and 3.3. Additionally, the determination of the pressure gradient and corresponding boundary conditions should be addressed in both approaches.*

   A clearer description of the Dirichlet boundary condition used have been added in section 2.2, with explicit pressure gradient and boundary conditions in section 2.4.

3. *For the numerical simulation sections, it should also be more clearly indicated which set of equations and boundary conditions for all variables were used for every numerical simulation. This will greatly help in the correct understanding and reproduction of the paper's results.*

   For each approach, the forcing functions are now summarized clearly at the end of sections 3.2 and 3.3.

4. *It would be beneficial to make the calculation input and output data available to the public to help test the other numerical models that may use the described approach for boundary conditions.*

   The input/output data for each simulation presented, along with the required analysis codes and the actual, in-solver, implementation of the Euler-Lagrange approach along with the eigenvalue/eigenvector solver are now accessible through Globus endpoint as elaborated in detail in the "Code and data availability" statement at the end of the paper. Until the paper is published, this data is available upon request.